# Safeguarding Drosophila female germ cell identity depends on an H3K9me3 mini domain guided by a ZAD zinc finger protein

Laura Shapiro-Kulnane, Micah Selengut, Helen K. Salz ⓘ *

Department of Genetics and Genome Sciences, Case Western Reserve University School of Medicine, Cleveland, Ohio, United States of America

* hks@case.edu

## Abstract

H3K9me3-based gene silencing is a conserved strategy for securing cell fate, but the mechanisms controlling lineage-specific installation of this epigenetic mark remain unclear. In *Drosophila*, H3K9 methylation plays an essential role in securing female germ cell fate by silencing lineage inappropriate *phf7* transcription. Thus, *phf7* regulation in the female germ-line provides a powerful system to dissect the molecular mechanism underlying H3K9me3 deposition onto protein coding genes. Here we used genetic studies to identify the essential cis-regulatory elements, finding that the sequences required for H3K9me3 deposition are conserved across *Drosophila* species. Transposable elements are also silenced by an H3K9me3-mediated mechanism. But our finding that *phf7* regulation does not require the dedicated piRNA pathway components, *piwi*, *aub*, *rhino*, *panx*, and *nxf2*, indicates that the mechanisms of H3K9me3 recruitment are distinct. Lastly, we discovered that an uncharacterized member of the zinc finger associated domain (ZAD) containing C2H2 zinc finger protein family, IDENTITY CRISIS (IDC; CG4936), is necessary for H3K9me3 deposition onto *phf7*. Loss of *idc* in germ cells interferes with *phf7* transcriptional regulation and H3K9me3 deposition, resulting in ectopic PHF7 protein expression. IDC's role is likely to be direct, as it localizes to a conserved domain within the *phf7* gene. Collectively, our findings support a model in which IDC guides sequence-specific establishment of an H3K9me3 mini domain, thereby preventing accidental female-to-male programming.

**Data Availability Statement:** All relevant data are within the manuscript and its Supporting Information files.

## Author summary

Tissue development and function relies on cells remembering their identity. A cell's identity is defined by the genes it expresses and those it does not. Recent work has shown that genes can be silenced by trimethylation of histone H3 lysine 9 (H3K9me3) marked chromatin, and that H3K9me3-mediated gene silencing is a vital strategy for securing cell fate. But there is very little information about how the machinery responsible for H3K9 methylation finds its target genes. Here we explore this issue using the *Drosophila* female germ-line where a mini domain of repressive H3K9me3 chromatin secures female germ cell fate by silencing *phf7*, a gene normally expressed in male germ cells. Transposable elements

**Funding:** This work was supported by the National Institutes of Health grant R01GM129478 to H.K.S. The funders had no role in study design, data collection and analysis, decision to publish, or preparation of the manuscript.

**Competing interests:** The authors have declared that no competing interests exist.

are also silenced by H3K9me3 mini domains, but we find that the proteins involved in this process are not required for *phf7* silencing. Instead, we find that silencing requires a previously uncharacterized protein, we have named IDENTITY CRISIS. Our work provides evidence that IDENTITY CRISIS directs the H3K9 methylation machinery to build a mini domain at the *phf7* locus. Our results shed new light into how cells safeguard their identity by silencing cell type inappropriate genes, and more specifically how these genes are identified by the silencing machinery.

## Introduction

Gene silencing is critical to establishing and maintaining cell fates. Once made, the decision to silence a gene is fortified by the acquisition of repressive histone modifications. While cell type specific epigenetic silencing is primarily associated with tri-methylation of histone H3 lysine 27 (H3K27me3)-marked chromatin, tissue specific genes can also be repressed by H3K9 methylation [1–4]. For example, in *S. pombe*, discrete H3K9me3 domains silence meiotic genes in vegetative cells (e.g., [5,6]). In *C. elegans*, H3K9 methylation silences inappropriate cell type specific genes, including germline genes, in somatic cells (e.g., [7,8]). In the *D. melanogaster* female germline, H3K9me3 silences male germline genes [9]. In the mouse, H3K9me3 silences germline genes during early embryonic development (e.g., [10,11]). Studies carried out in mammalian tissue culture systems further identify H3K9me3-mediated gene silencing as a conserved and vital strategy for maintaining cell fates in a wide range of tissues (e.g., [12–23]). However, the molecular mechanisms controlling tissue specific installation of this epigenetic modification onto protein-coding genes are largely unknown.

The repressive H3K9me3 histone modification has well characterized roles in constitutive heterochromatin formation, and the transcriptional silencing of repetitive DNA elements such as satellite repeats and transposable elements (TEs) [24–26]. These studies identified two mechanisms of H3K9me3 recruitment. One mechanism involves small RNAs that guide localization through a complementary base pairing mechanism. In *Drosophila* and mammals, for example, the PIWI-associated small RNAs (piRNAs) guide the H3K9me3 silencing machinery to TEs [27]. A second mechanism involves sequence specific DNA binding proteins. In mammals, for example, H3K9me3 can be guided to TEs by members of the vertebrate specific Kruppel-associated box zinc finger (KRAB-ZNF) family of DNA binding proteins [28–31]. An analogous mechanism might exist in *Drosophila*, as two zinc finger proteins, KIPFERL and SMALL OVARY, were recently shown to have roles in TE silencing [32–35]. Whether installation of this epigenetic modification onto protein-coding genes employs similar mechanisms remains unclear.

In *Drosophila*, H3K9 methylation plays an essential role in securing female germ cell fate by silencing lineage inappropriate <u>PHD finger protein 7</u> (*phf7*) transcription [9,36]. Thus, *phf7* regulation in the female germline provides a powerful system to investigate how the H3K9me3 silencing mark is installed at protein-coding genes. *phf7* encodes a predicted chromatin reader, first identified in a screen for genes expressed in male but not female embryonic germ cells [37]. Curiously, *phf7* is not essential for male fertility, but it is critical that female germ cells not express the PHF7 protein. Forced expression of PHF7 activates a toxic gene expression program enriched for genes usually restricted to the male germline [36]. Even though PHF7 protein is limited to male germ cells, *phf7* is transcribed in both male and female germ cells. Sex specificity is achieved by alternative transcription start sites (TSS; **Fig 1A**). In testes, transcription from the upstream TSS (TSS1) produces a long mRNA isoform that makes protein.

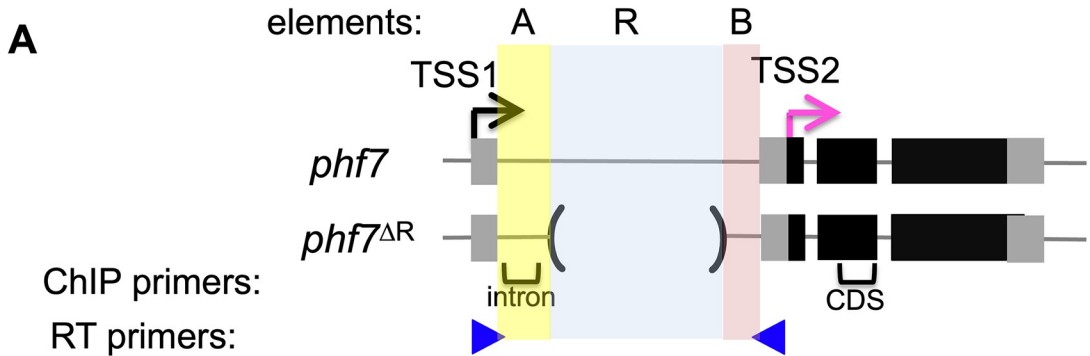

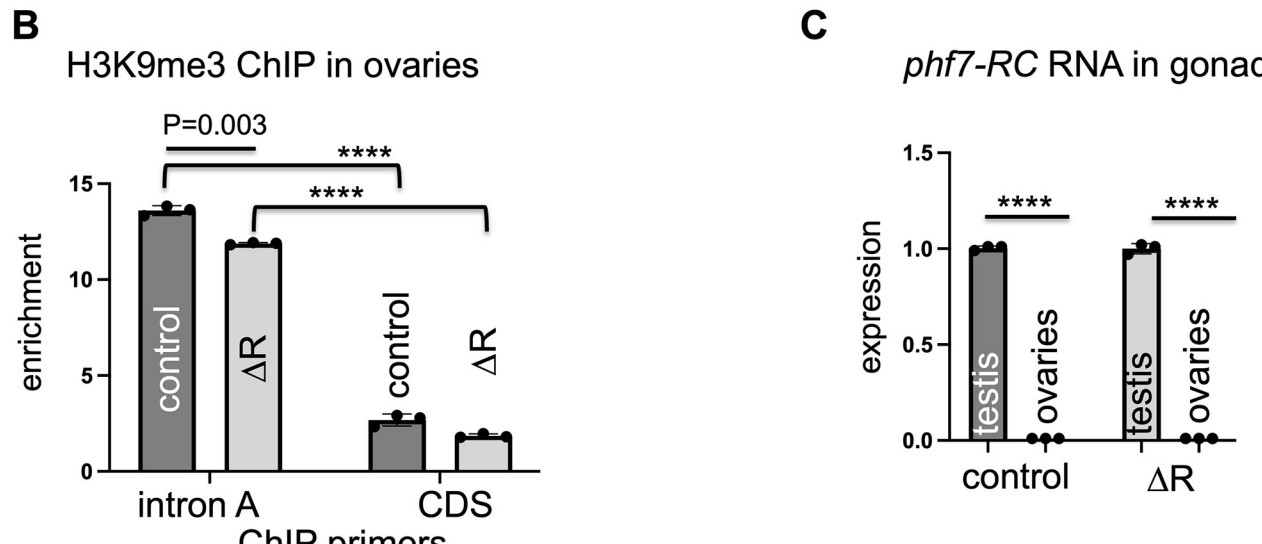

**Fig 1. H3K9me3 deposition and sex specific transcriptional regulation are maintained in *phf7^ΔR* ovaries.** (A) Diagram of the *phf7^+* and *phf7^ΔR* alleles. Exons are represented by boxes, flanking DNA, and introns by lines, black boxes are coding sequences, and grey boxes are untranslated regions (UTRs). The sequence elements A, R, and B location within the first intron are shaded in yellow, blue, and pink. In ovaries, transcription initiates from TSS2 (pink arrow), whereas an H3K9me3-mediated mechanism represses transcription from TSS1 (black arrow). Locations of primers for ChIP-qPCR and RT-qPCR are indicated by brackets & blue arrowheads. **(B)** H3K9me3 accumulation at the endogenous *phf7* locus in control (*y^1 w^1*) and mutant ovaries. ChIP-qPCR measured H3K9me3 signal at the *phf7* locus. ChIP to input enrichment is normalized to *rp49*. **(C)** Expression of the testis specific *phf7* transcript (*phf7-RC)* in control (*y^1 w^1*) and mutant ovaries. Expression is measured by RT-qPCR and is shown as the fold change relative to the testis. Expression is normalized to the total level of *phf7*. For both RT-qPCR and ChIP-qPCR experiments. Error bars represent the standard error of the mean (SD) of three biological replicates. A two-tailed Student's t-test estimates statistical significance where ****$p < 0.0001$.

In ovaries, an H3K9me3 mini domain prevents the selection of TSS1. Instead, transcription initiates from the downstream TSS (TSS2) to produce a short mRNA isoform that is not translated and has no function. Of the three *Drosophila* enzymes known to methylate H3K9, only SETDB1 has a specific and nonredundant role in repressing *phf7* [9]. Germ cell specific loss of SETDB1, its binding partner ATF7IP, or the H3K9 reader HP1a produced ovarian germ cell tumors that inappropriately express the PHF7 protein. Notably, derepression results from losing the H3K9me3 mini domain. While these results establish the H3K9me3 mini domain controls *phf7* transcription, the mechanisms controlling installation of this epigenetic modification remains unclear.

We address the mechanism of targeted H3K9me3 deposition onto *phf7* by identifying essential cis-regulatory elements and trans-acting factors. We establish that the required cis-regulatory sequences are conserved across *Drosophila* species and find that the mechanism governing *phf7* regulation is different from what has been described for piRNA-guided H3K9me3 deposition on TEs. Lastly, we discover that repression depends on the previously unknown gene, *CG4936*, that we have named *identity crisis (idc)*. *idc* encodes a zinc finger associated domain (ZAD) containing C2H2 zinc finger protein. Notably, loss of *idc* in germ cells interferes with *phf7* repression by reducing H3K9me3 deposition. Together with the observation that IDC localizes to a conserved region within the *phf7* gene, our analysis supports a model in which IDC guides H3K9me3 installation, thereby preventing accidental female-to-male programming.

## Results

### Identification of cis-regulatory elements required for H3K9me3 deposition

H3K9me3 accumulates over a three kb region within the *phf7* gene [9]. This discrete peak covers the male TSS1, the first male-specific, non-coding exon, and most of the first intron. Interestingly, the intron contains seven copies of ~250 bp sequence not found anywhere else in the genome (element R; **Figs 1A** and **S1**). Repetitive elements can regulate gene expression in *cis* by serving directly as H3K9me3 nucleation sites or in *trans* by encoding small RNAs that serve as guides via a base pairing mechanism. We, therefore, hypothesized that the repeats play a role in H3K9me3 deposition and silencing of TSS1. To test this idea, we used CRISPR editing to produce animals deleted for the repeats in the endogenous locus (*phf7^{ΔR}*) (**Fig 1A**). We were surprised to discover that homozygous *phf7^{ΔR}* animals are viable and fertile, suggesting that the repeats may not be essential for repression. To evaluate the impact of deleting the repeats in more detail, we measured H3K9me3 accumulation in wild-type and *phf7^{ΔR}* homozygous ovaries using chromatin immunoprecipitation followed by quantitative PCR (ChIP-qPCR). In wild-type (control) ovaries, sequences in the intron positioned 319 bp downstream of TSS1 (within intronic element A) are significantly enriched in the H3K9me3 modification when compared to the region within the coding sequence (CDS; $p<0.0001$) (**Fig 1B**). Significant H3K9me3 enrichment within element A compared to the CDS region ($p<0.0001$) was also observed in *phf7^{ΔR}* homozygous ovaries. Interestingly, there was less H3K9me3 accumulation within element A in *phf7^{ΔR}* than in controls ($p = 0.003$), raising the possibility that sex specific transcriptional regulation might be disrupted. In testes, transcription from the upstream TSS (TSS1) produces a long mRNA isoform, called *phf7-RC*. We therefore used RT-qPCR to assay for testis specific *phf7-RC* expression levels in wild-type and *phf7^{ΔR}* gonads. Contrary to our expectations, we found that *phf7^{ΔR}* did not disrupt sex specific transcription: As in wild-type gonads, *phf7-RC* expression was detected in *phf7^{ΔR}* homozygous testis but not ovaries (**Fig 1C**). These results indicate that the repeats are not essential for H3K9me3-mediated repression.

To complement these studies, we used a transgenic approach to identify the sequences capable of promoting H3K9me3 when inserted into a heterologous genomic location on the 3rd chromosome using site specific integration. We first created a transgene, that mimics the 5' end of the *phf7^{ΔR}* mutant allele, extending from the first male specific exon to the beginning of the open reading frame in exon 2 (**S2 Fig**). To assay for H3K9me3 accumulation on the ΔR transgene, but not at the endogenous locus, we crossed each line into a *phf7^{Δ13}* background. *phf7^{Δ13}* is a 1.81 kb intragenic deletion allele, which allows us to measure the level of H3K9me3 deposition by ChIP-qPCR at a region that is present in the transgenes but absent in *phf7^{Δ13}* (intronic element B, **Fig 2A**). As expected, we found that these sequences included in the ΔR

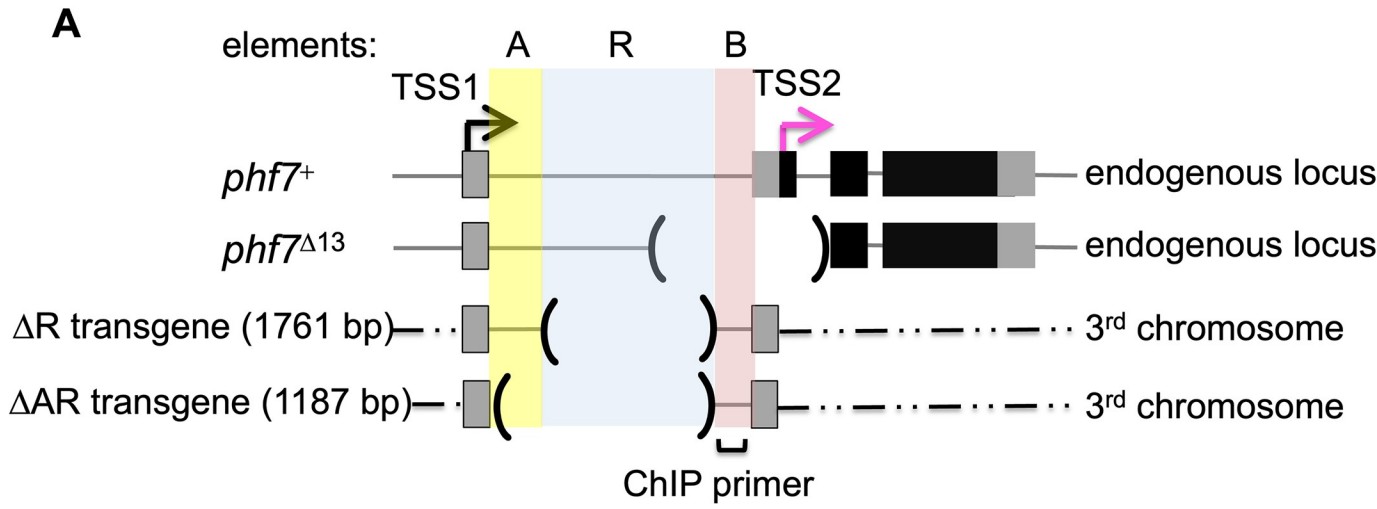

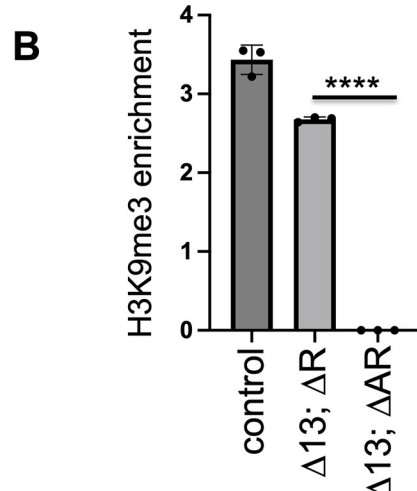

**Fig 2. Non-coding sequences within the first intron are required for H3K9me3 deposition.** (**A**) Diagram of *phf7+*, *phf7^Δ13^*, and the ΔR and ΔAR chromatin transgenic reporter lines inserted into the same 3^rd^ chromosomal site. (**B**) H3K9me3 accumulation on the transgenes. ChIP-qPCR measured signal, and the ChIP to input enrichment, normalized to *rp49*. Error bars represent the standard error of the mean (SD) of three biological replicates. A two-tailed Student's t-test estimates statistical significance where ****$p<0.0001$.

transgene promotes H3K9me3 accumulation. (**Fig 2A and 2B**). These data reinforce our conclusion that the tandem repeats (element R) are not required for H3K9me3 accumulation. In contrast, we found that H3K9me3 did not accumulate on a second transgene in which both elements A and R were removed (**Fig 2A and 2B**). We therefore conclude that the sequences that remain, including the first exon and region B, are not sufficient for H3K9me3 recruitment. These results also establish that element A contains cis-regulatory determinants required for H3K9me3 recruitment. It remains to be determined whether the element A determinants are sufficient for H3K9me3 recruitment or function redundantly with sequences within element R.

## Conservation of non-coding cis-regulatory elements

Cis-regulatory elements are often conserved. Thus, we might expect the essential sequences within element A to be retained in evolution. To test this prediction, we chose to examine the

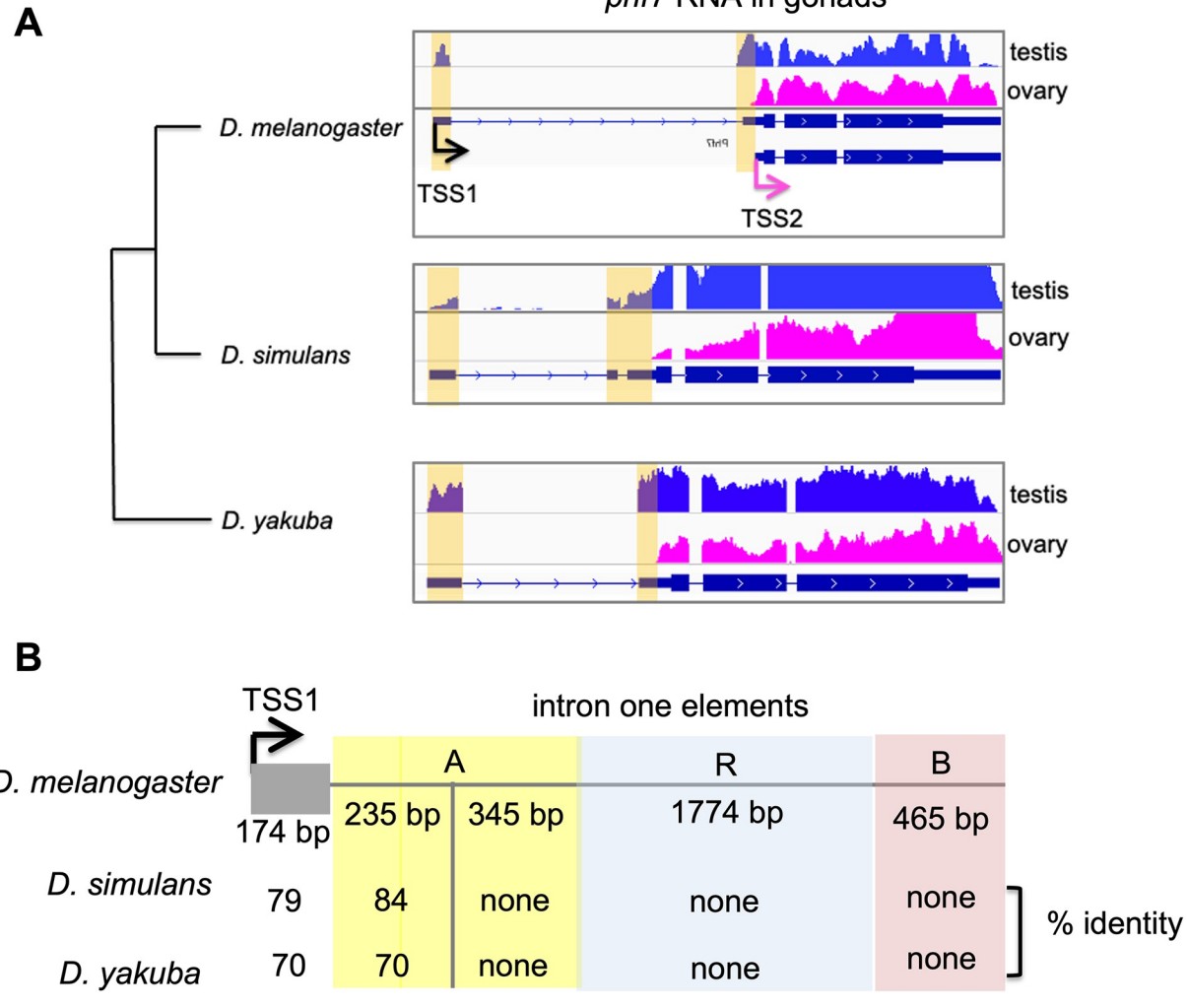

**Fig 3. Sex specific transcriptional regulation is conserved. (A)** Genome browser views of RNA-seq data aligned to the *phf7* locus from *D. melanogaster*, *D. simulans*, and *D. yakuba* ovaries and testis. **(B)** The first exon and a portion of the first intron are conserved. Pairwise alignments between the first exon and the adjacent intron were identified using the Basic Local Alignment Search Tool (BLASTn) available at (https://blast.ncbi.nlm.nih.gov). Percent identity to the Drosophila melanogaster to the first male specific exon and element A is shown. The sequence alignments are presented in S3 Fig.

level of conservation between *D. melanogaster*, *D. simulans*, and *D. yakuba*. Although *D. simulans* and *D. yakuba* are separated from *D. melanogaster* by 5–15 million years, the *phf7* intron/exon structure is conserved (**Fig 3A**). Furthermore, RNA-seq data obtained from wild-type testis and ovaries [38,39] shows that *phf7* expression is sex specific in *D. simulans* and *D. yakuba* (**Fig 3A**). Together these observations suggest that sex-specific transcriptional regulation through alternative TSS selection is conserved between *D. melanogaster*, *D. simulans*, and *D. yakuba*. When we compared the level of sequence conservation in the first intron between *D. melanogaster*, *D. simulans*, and *D. yakuba*, we found that only the first 235 bp of intron element A was conserved (**Figs 3B and S3**). Interestingly, the first non-coding 174 bp exon also displays a high level of conservation, suggesting that additional cis-regulatory elements may be present within the first non-coding exon (**Figs 3B and S3**)

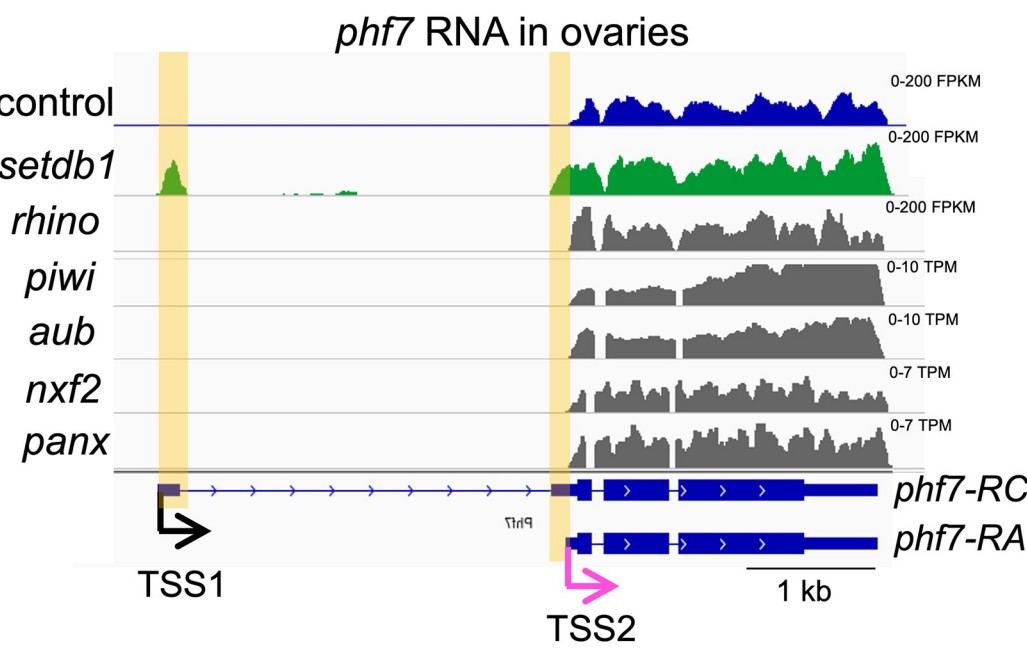

**Fig 4. Depletion of *setdb1*, but not the dedicated piRNA-pathway components, leads to female-to-male reprogramming at the *phf7* locus.** Genome browser views of the *phf7* locus. Tracks show RNA-seq reads from control and mutant ovaries aligned to the Drosophila genome (UCSC dm6). The screenshot is reversed so that the 5' end of the gene is on the left. The RNA-seq reads that are unique to *setdb1* mutant ovaries are highlighted. In wild-type control ovaries, only the *phf7-RA* transcript is detected (TSS2, pink arrow), whereas loss of *setdb1* leads to ectopic expression of the testis specific *phf7-RC* isoform (TSS1, black arrow). In contrast, mutations in the *rhino*, *piwi*, *aub*, *nxf2* and *panx* genes do not disrupt sex specific *phf7* transcription. The complete mutant genotypes and accession numbers of the RNA-seq data sets are listed in S3 Table.

## Core components of the piRNA pathway are not required for *phf7* sex specific transcriptional control

Prior studies have shown that SETDB1 is required for both TE and *phf7* silencing in germ cells [9,40–42]. Although there are no recognizable TE sequences at *phf7*, the piRNA pathway may nevertheless play a role in repressing *phf7*. To test this possibility, we analyzed published RNA-seq data sets from ovaries carrying mutations in genes encoding the dedicated piRNA pathway components, *piwi*, *aubergine (aub)*, *rhino*, *panoramix (panx*, also known as *silenceo)*, and *nuclear RNA export factor 2 (nxf2)* [43–45]. Loss of each component causes the piRNA pathway to collapse. *piwi*, *aub* and *rhino* are essential for piRNA biogenesis [42, 43,46]. *panx* and *nxf2*, while dispensable for piRNA biogenesis, are necessary for SETDB1 to deposit H3K9me3 marks onto TEs [45, 47–51]. In agreement with our prior studies [9], only the shorter *phf7-RA* transcript is detectable in wild-type ovaries, but the longer testis specific *phf7-RC* transcript is present in *setdb1* mutant ovaries (**Fig 4**). In contrast to the *setdb1* mutant ovaries, *piwi*, *aub*, *rhino*, *panx*, and *nxf2* mutant ovaries only express the shorter *phf7-RA* transcript, indicating that transcriptional regulation is not disrupted. Based on these findings we conclude that the mechanisms controlling H3K9me3 deposition onto *phf7* and TEs are distinct.

## The ZAD-ZNF protein IDC is required for *phf7* sex specific transcriptional control

In mammalian cells, SETDB1 can be recruited to its targets by members of the large KRAB-zinc finger family of sequence specific DNA binding proteins [28–30]. The KRAB family is

confined to mammals. It has been hypothesized that members of the insect specific ZAD-ZNF family might be functional analogues [52–54]. We recently tested the function of 68 of the 93 ZAD-ZNF encoding genes in female germ cells by performing an RNAi screen [55]. This screen identified eight ZAD-ZNF genes required for oogenesis. Here, we focus on CG4936 which we name *identity crisis* (*idc*). *idc* encodes a 521 amino acid (aa) protein that contains an N-terminal ZAD (zinc finger associated domain, 22–95 aa), an unstructured linker region, and a C-terminal domain that includes an array of 5 C2H2 zinc fingers (386–491 aa). Studies of other ZAD-ZNF proteins suggest that the ZAD mediates protein-protein interactions and the C2H2 zinc fingers bind DNA [56–60]. Interestingly, the knockdown phenotype of IDC suggests a defect prior to oocyte specification [55]. Because this phenotype is reminiscent of the germ cell defects caused by ectopic PHF7 expression, we hypothesize that IDC is required for *phf7* repression.

To investigate the possibility that loss of *idc* in female germ cells disrupts *phf7* repression, we first used RT-qPCR to assay for the presence of the testis specific *phf7-RC* transcript. Germ cell specific knock down (GLKD) was achieved by expressing a germline optimized inducible RNA interference (RNAi) transgene with the germ cell specific *nos-GAL4* driver. We demonstrated knockdown efficiency by showing that *idc* GLKD significantly reduces IDC protein levels in female germ cells, but not in the surrounding somatic cells (**S4 Fig**). To rule out indirect effects arising from RNAi knockdown, we also knocked down *white*, which is not expressed in germ cells, as a negative control. Using primer pairs capable of detecting *phf7-RC*, we found that in contrast to control ovaries, *phf7-RC* is detectable in *idc* GLKD mutant ovaries (**Fig 5A**). We then asked whether ectopic *phf7-RC* expression correlated with ectopic PHF7 protein expression. Previous work has shown that a BAC transgene that encodes an N-terminally HA-tagged PHF7 protein (HA-PHF7) is a faithful reporter of the sex specific protein expression pattern of PHF7 [37,61]. We found that, in contrast to wild-type ovaries, the HA-PHF7 protein is detectable in the cytoplasm and the nucleus of the *idc GLKD* mutant ovaries (**Fig 5B and 5C**). We conclude that IDC is required for *phf7* regulation.

To test whether *idc* plays a role in controlling H3K9me3 deposition, we compared the amount of H3K9me3 accumulation within element A in wild-type and mutant ovaries using

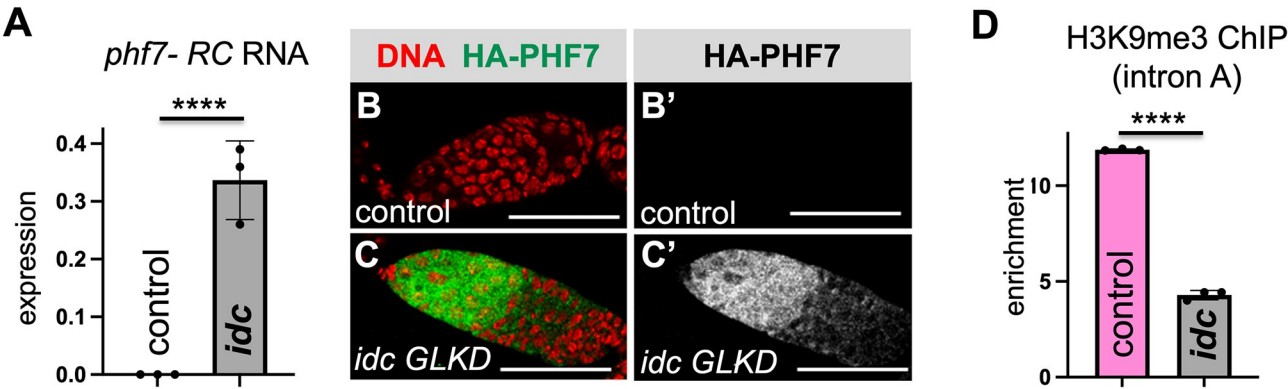

**Fig 5. *idc* germ cell specific knock-down disrupts *phf7* regulation. (A)** *idc* GLKD germ cells (*nos>idc-RNAi*) express the testis *phf7-RC* transcript. RT-qPCR measures expression normalized to the total level of *phf7* in RNA extracted from control (*nos>white-RNAi*) and *idc* GLKD (*nos>idc-RNAi*) ovaries. The histograms show the mean ± SD of three biological replicates. **(B, C)** *idc* GLKD germ cells inappropriately express the PHF7 protein. Confocal images of ovaries dissected from control (*nos>white-RNAi*) and mutant (*nos>idc-RNAi*) females carrying an HA-PHF7 transgene and stained for HA (green, white in B' and C') and DNA (red). Scale bar 50μm. **(D)** H3K9me3 accumulation at the endogenous *phf7* locus is reduced in *idc* GLKD ovaries. ChIP-qPCR measured H3K9me3 signal at the *phf7* locus (primer in region A) in control (*nos>white-RNAi*) and mutant (*nos>idc-RNAi*) ovaries. ChIP to input enrichment is normalized to *rp49*. The histogram shows the mean ± SD of three biological replicates. A two-tailed Student's t-test estimates statistical significance where ****p<0.0001.

ChIP-qPCR. We found that H3K9me3 was significantly reduced in *idc* GLKD ovaries compared to control ovaries (*p*<0.0001; **Fig 5D**). Based on these studies, we conclude that IDC regulates *phf7* transcription by promoting H3K9me3 deposition.

## IDC is a nuclear protein that associates with DNA in ovaries

Published modENCODE RNA-seq data sets indicates that the *idc* RNA is broadly expressed throughout development [62]. To examine IDC protein expression in the ovary, we used a genomic fosmid transgene that encodes a C-terminally GFP-tagged IDC protein (IDC-GFP). We inferred that the GFP tag does not interfere with *idc* function because the *idc-GFP* rescues the *idc¹* lethal phenotype (see Materials and Methods). Each ovary is comprised of 16–20 ovarioles. Each ovariole contains germ cells spanning the range of maturity from germline stem cells (GSCs) at the anterior end to fully mature eggs at the posterior end [63]. We observed IDC-GFP in the somatic cells, the nurse cells and the oocyte (**Fig 6A**). Notably, the IDC-GFP protein is nuclear. Furthermore, IDC-GFP is tightly associated with the nurse cell polytene chromosomes, consistent with its presumed DNA binding activity (**Fig 6B**).

In the germarium, at the anterior end of the ovariole, we observed prominent IDC-GFP staining in the nucleus of only 3 to 6 germ cells (**Fig 6A**). This expression pattern is like that described for the female specific sex determination protein SEX-LETHAL (SXL), which is also required for *phf7* H3K9me3 silencing [9, 64,65]. Indeed, co-staining experiments reveal that SXL accumulates in the cytoplasm of all IDC-GFP expressing germ cells (**Fig 6C**). Studies have shown that SXL accumulates in GSCs and their immediate daughter cells. When GSCs divide, the daughter cells at the tip remain a GSC, while the more posterior daughter cells, called a cystoblast (CB), express the BAG OF MARBLES protein (BAM). In agreement with published studies, BAM expression, assessed with a transgene that encodes a C-terminally HA-tagged BAM protein expressed from a *bam* promoter is readily detectable in the cytoplasm of just a few germ cells [66]. Co-staining experiments reveal that IDC-GFP is expressed in the GSCs where BAM-HA is not expressed as well as in the adjacent two to three cells where BAM-HA is detectable (**Fig 6D**). Together, these observations indicate that the nuclear IDC protein is broadly expressed, with prominent expression in the GSCs and their immediate daughter cells.

## IDC binds to sequences within the male specific exon

IDC contains five zinc fingers suggesting that it has sequence specific DNA binding activity. We speculated that IDC might promote H3K9me3 deposition by binding to the *phf7* locus. To test this prediction by ChIP-qPCR, we designed primers along the length of the conserved noncoding sequences in the first exon and the intron. These experiments revealed that the IDC-GFP tagged protein associates with chromatin within the first non-coding exon, but not outside of it (**Fig 7A**). Collectively, our studies indicate that IDC regulates *phf7* directly by serving as an H3K9me3 guidance factor.

## Discussion

SETDB1-controlled H3K9 methylation plays an essential role in securing female germ cell fate by silencing lineage-inappropriate *phf7* transcription [9,36]. SETDB1 is also required for TE silencing, where a piRNA-guided mechanism guides H3K9me3 deposition [40–42]. In this work, we establish that *phf7* is silenced by a piRNA-independent mechanism and discover that regulation depends on IDC, an uncharacterized member of the ZAD-ZNF family of DNA binding proteins. Regulation appears direct, as IDC is required for H3K9me3 deposition and localizes to the conserved first exon of *phf7* in ovarian extracts. Collectively, our data establish that the sequence specific DNA binding protein IDC directs the H3K9 methylation machinery

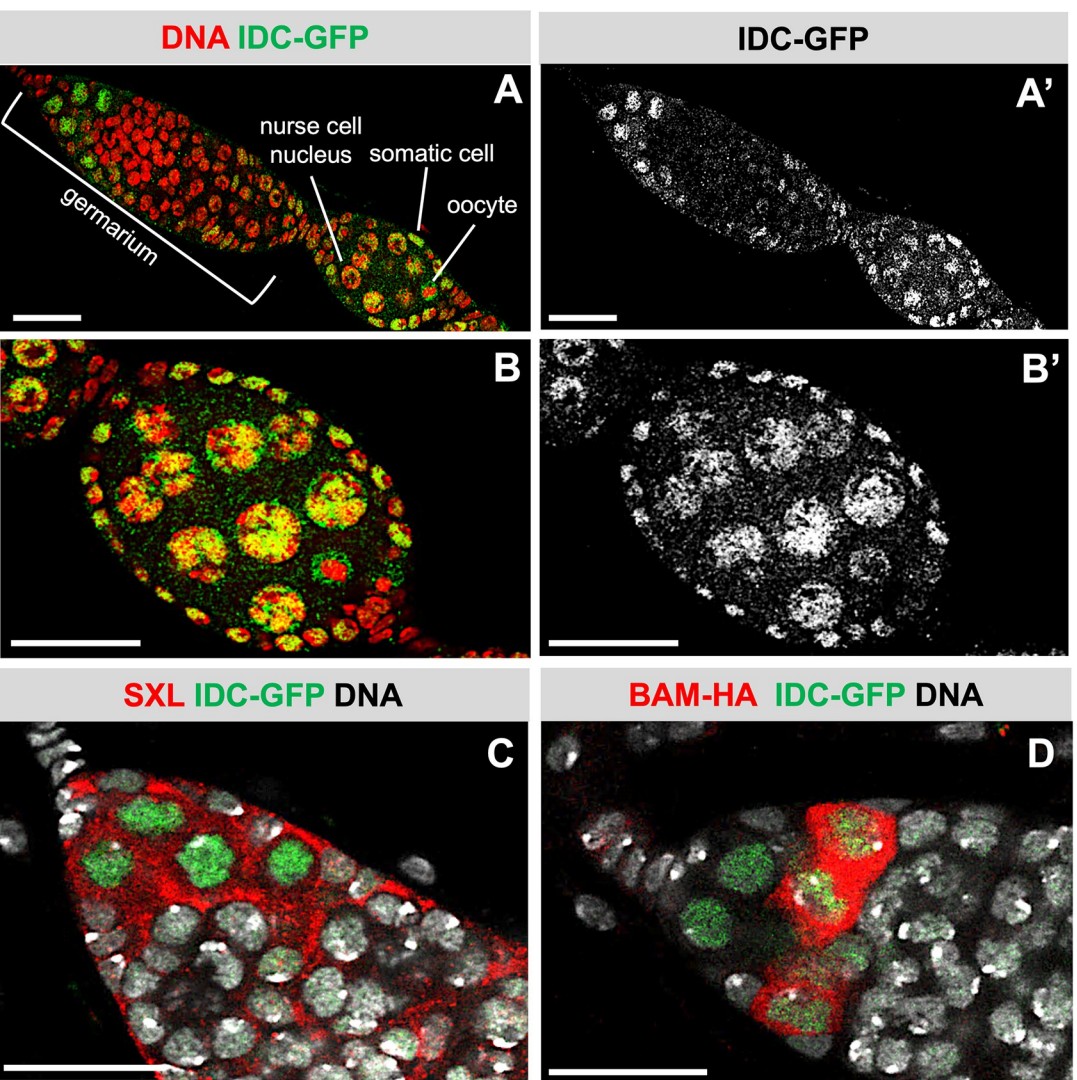

**Fig 6. IDC is nuclear protein expressed in both somatic and germline cells. (A)** Confocal image of an ovariole dissected from a female carrying the IDC-GFP rescue transgene stained for GFP (green, white in A') and DNA (red). The bracket indicates the germarium located at the anterior end of the ovariole. Scale bar 25μm. **(B)** Confocal image of an early-stage egg chamber dissected from a female carrying the IDC-GFP transgene stained for GFP (green, white in B') and DNA (red). Scale bar 25μm. **(C)** Confocal image of the anterior end of a germarium dissected from a female expressing the IDC-GFP fusion protein and co-stained for GFP (green), DNA (white), and SXL (red). Scale bar 25 μm. **(D)** Confocal image of the anterior end of a germarium dissected from a female expressing the IDC-GFP and the BAM-HA fusion proteins co-stained for GFP (green), DNA (white), and HA (red). Scale bar 25μm.

to build a silencing domain at the *phf7* locus, thereby preventing accidental female-to-male reprogramming (**Fig 7B**). In addition to extending our understanding of how female germ cell fate is maintained, our studies provide the first example of a ZAD-ZNF protein guiding H3K9me3-mediated gene silencing.

Although our work is consistent with a simple model in which the SETDB1 H3K9me3 methyltransferase is recruited to *phf7* by IDC, the mechanism by which IDC guides the methylation machinery to *phf7* remains an open question. For example, it remains unclear whether recruitment is direct, as our attempts to co-immunoprecipitate SETDB1 and IDC were unsuccessful. Furthermore, while we establish that IDC is required for H3K9me3 recruitment, our

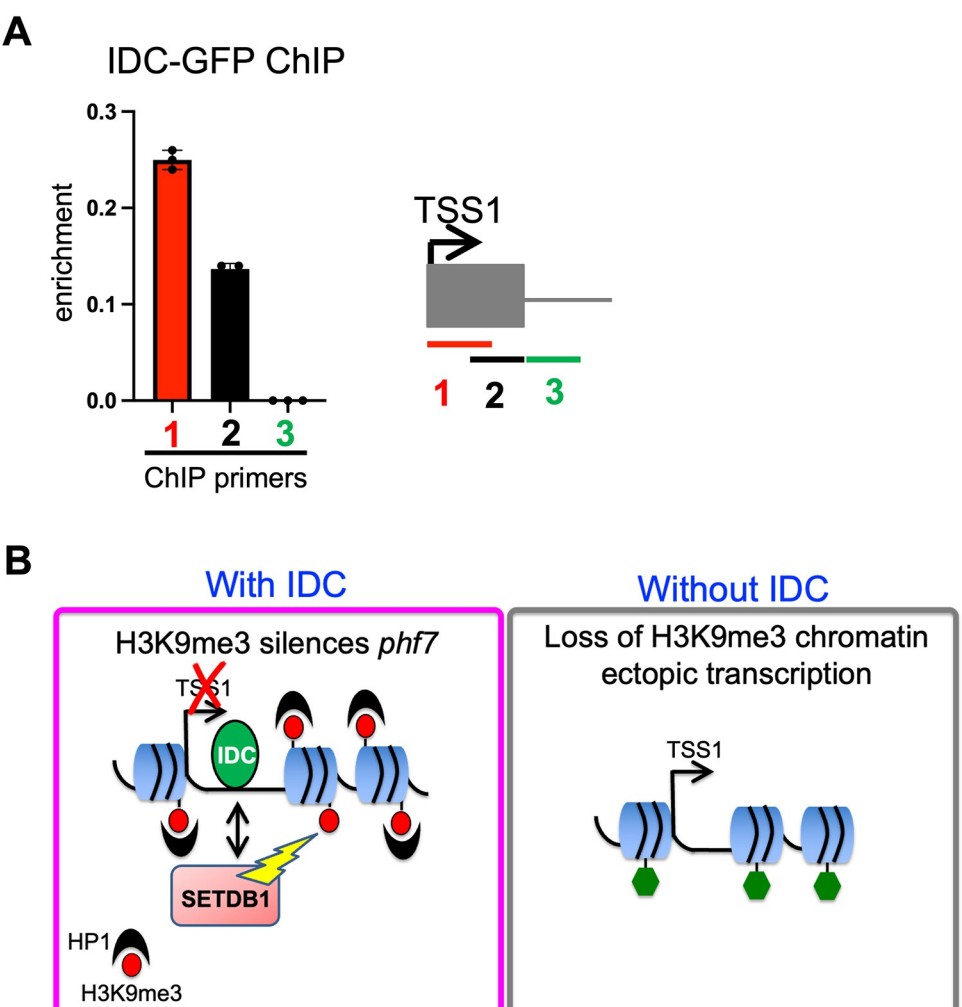

**Fig 7. IDC binds to *phf7*.** (A) ChIP-qPCR shows that IDC binds to the *phf7* first exon. Left: The histogram shows the mean ± SD of three biological replicates. Right: Cartoon showing the location of the primers used for ChIP-qPCR. **(B)** Model for H3K9me3-mediated silencing of *phf7*. In female germ cells, IDC binds to *phf7*, directing H3K9me3 deposition by SETDB1 H3K9 methyltransferase. HP1 binds to H3K9me3, resulting in transcriptional silencing. In germ cells lacking IDC, the dissolution of the H3K9me3 domain correlates with ectopic testis-specific *phf7-RC* transcription and PHF7 protein expression. Ectopic PHF7 activates a toxic gene expression program enriched for genes usually restricted to the male germline.

chromatin transgenic reporter assays show that the region to which it binds, the conserved first exon, is not sufficient. This observation, together with our identification of a second conserved cis-regulatory element within the adjoining intron invites speculation that IDC works in conjunction with other sequence-specific recruitment factors. One attractive contender is *stonewall (stwl)*. STWL is a heterochromatin-associated protein that acts as a transcriptional repressor *in vitro* [67], associates with SETDB1 in yeast [68] and localizes to the *phf7* locus in S2 cells [69]. Importantly, loss of *stwl* in ovaries leads to the inappropriate expression of the testis *phf7* transcript [69]. Future studies focused on the rules that govern female-specificity may reveal a general mechanism for context-dependent establishment of H3K9me3 silencing domains.

We have previously established that the H3K9me3 reader protein, HP1a, is essential for *phf7* silencing [9]. A requirement for HP1a is not surprising, as HP1a drives chromatin

compaction and transcriptional silencing [70]. HP1a also recruits H3K9 methyltransferases, enabling the spreading of the H3K9me3 repressive domain through a positive feedback mechanism. The spread of H3K9me3 over the three kb region within the *phf7* gene likely occurs by an analogous mechanism. At the *phf7* locus, however, the H3K9me3 domain does not extend into the open reading frame or the neighboring genes. Therefore, more work is needed to understand what stops the spreading of this repressive chromatin domain.

The SETDB1-controlled H3K9 methylation silencing pathway in female germ cells is not restricted to *phf7*. Genome-wide H3K9me3 profiling in wild-type and mutant ovaries has shown that SETDB1 silences two classes of protein-coding genes. One type includes genes usually expressed in the testis [9]. A second class includes genes that are typically expressed in undifferentiated female germ cells but are silenced once the oocyte is specified, such as *ribosomal protein S19b* [71]. These few examples of context dependent H3K9me3 gene silencing suggest a finely tuned guidance mechanism in the female germline. It will be interesting to explore whether other members of the ZAD-ZNF gene family serve as H3K9me3 guidance factors.

In summary, by focusing on a single biologically relevant gene, we discovered a putative DNA binding protein that guides the installation of a H3K9me3 repressive domain onto a protein-coding gene. These findings are reminiscent of how TEs can be silenced in mammals wherein members of the KRAB-ZNF protein family recruit SETDB1 to establish epigenetic repression [28–31]. Interestingly, the KRAB-ZNFs are vertebrate specific, and the ZAD-ZNFs are insect specific. Yet, both gene families exhibit similar patterns of species-specific gene expansion and diversification. These observations have fueled the speculation that ZAD-ZNFs and KRAB-ZNFs perform similar functions [52–54]. In fact, IDC's closest human relative is ZNF75D, a KRAB-ZNF protein of unknown function. Whether this or other members of the KRAB-ZNF protein family are required for tissue specific H3K9me3 gene silencing will be an exciting area of further exploration.

## Materials and methods

### *Drosophila* stocks and culture conditions

Fly strains were maintained on standard molasses food at 25˚C unless otherwise noted. Wild-type control flies were from the lab $y^1 w^1$ stock. Stocks used to report on protein gene activity include the HA-tagged *phf7* transgene, *PBac{3XHA-PHF7}* [37]; the GFP-tagged *idc* transgene, *FlyFos{fTRG00376.sfGFP-TVPTBF}* (VDRC #318556) described in [72] and the HA-tagged *bam* transgene, *P{Bam::HA}* [66]. *phf7*$^{Δ13}$, *phf7*$^{ΔR}$, the *idc*$^1$ null allele, and the ΔR and ΔAR chromatin reporters were generated for this study, as described below.

Crosses for knock-down studies were set up at 29˚C, and adults were aged 3–5 days before gonad dissection. Knock-down studies were carried out with the following germline-optimized lines generated by the *Drosophila* Transgenic RNAi Project [73]: *idc-P{TRiP. HMC05569}* (BDSC #64550), and the control *w-P{TRiP.GL00094}* (BDSC #35573). Although off-target effects remain a concern for RNAi-induced knockdown studies, neither RNAi line is predicted to have off-target activity (https://www.flyrnai.org/up-torr/UptorrFly.jsp). Expression of the UAS-RNAi transgenic lines was driven by the third chromosome *P{GAL4:: VP16-nos.UTR}* insertion (BDSC #4937), which is strongly expressed in germ cells.

### Generation of the *phf7*$^{ΔR}$ allele

We generated the *phf7*$^{ΔR}$ allele in two steps. First, CRISPR was used to replace the repeats within the first intron with a 3XP3::DsRed cassette by Rainbow Transgenic Flies, Inc. The deletion was generated with the following guide RNAs: agttaaaaaaaatcaatcgatgg and

cgcagcgattgaatgttaatggg. 1 kb homology arms were generated through PCR and cloned into the pHD-dsRed-attP (Addgene #51019) [74]. Guide RNAs and the donor vector were co-injected into *nos-Cas9* embryos by Rainbow Transgenic Flies. Flies were screened for DsRed expression in the eyes, and sequence verified for accuracy. *phf7*$^{\Delta R}$ was generated by removing the DsRed cassette. Homozygous *phf7*$^{\Delta R+dsRED}$ females were crossed to a Cre-expressing fly line (BDSC #1092), and the male progeny screened for the loss of dsRED expression in the eyes, followed by sequence confirmation of precise tag excision.

## Generation of the *phf7*$^{\Delta 13}$ allele

The *phf7*$^{\Delta 13}$ allele was generated by P-element mediated imprecise excision as follows: females homozygous for the P-element insertion P{EPgy2}Phf7$^{EY03023}$ (BDSC #15894) were crossed with the Δ2–3 transposase expressing line (BDSC #3664). Potential excision lines were established from the male progeny of the F1 females that had lost the *white*$^+$ marker carried by the P element. Stocks that carried deletions were identified by PCR. The exact breakpoints were determined by DNA sequencing the PCR amplified DNA fragments.

## Generation of the ΔR and ΔAR chromatin transgenic reporter lines

Constructs were generated by VectorBuilder's custom cloning services (https://en.vectorbuilder.com). The *phf7* sequences, described in **S2 Fig**, were inserted into their "user-defined promoter" modification of the pUASTattB expression vector. The vectors were sent to Rainbow Transgenic Flies Inc. for *phi-C31* catalyzed integration into the 65B2 PBac{y[+]-attP-3B}VK00033 site. Transgenic flies were sequence verified for accuracy.

## Generation of the *idc*$^1$ null allele and rescue by the GFP-tagged *idc* transgene

The *idc*$^1$ allele was generated by *in vivo* CRISPR mutagenesis [75]. Females expressing two sgRNAs under control of the GAL4/UAS system, P{HD_CFD00598} (VDRC #341525), were crossed at 25°C to M{vas-Cas9}; P{GAL4::VP16-nos.UTR} (BDSC #55821 + #4937) males. The *vas-Cas9; nos>gRNA* male offspring were crossed to a third chromosome balancer line to isolate and balance each putative *idc* allele in the next generation. *idc* alleles were identified by the failure to complement Df(3R)ED6027 (BDSC #9479) and checked for the presence of the predicted indel by PCR.

We found that *idc*$^1$/Df(3R)ED6027 animals do not survive to adulthood (n>100). To verify that the GFP-tagged *idc* transgene, PBac{fTRG00376.sfGFP-TVPTBF} (VDRC #318556), rescues *idc*$^1$, we first made the double mutant *idc*$^1$, *idc-GFP*. To test for rescue, we crossed *idc*$^1$, *idc-GFP/TM3* to Df(3R)ED6027/TM3 males. This cross yielded 84% (31/37) of the expected *idc*$^1$, *idc-GFP/ Df(3R)ED6027* progeny, all of which were fertile.

## qRT-PCR and data analysis

RNA was extracted from dissected gonads using TRIzol (Thermo Fisher, cat# 15596026). Quantity and quality were measured using a NanoDrop ND-1000 spectrophotometer. The RNA was treated with DNase RQ1 (Promega, cat# M6101). cDNA was generated by reverse transcription using a SuperScript First-Strand Synthesis System for RT-PCR kit (Thermo Fisher, cat# 11904018) using random hexamers. qPCR was performed using *Power* SYBR Green PCR Master Mix (Thermo Fisher, cat# 4368706) with an Applied Biosystems QuantStudio 3 Real-Time PCR system. PCR steps were as follows: 95°C for 10 minutes followed by 40 cycles of 95°C for 15 seconds and 60°C for 1 minute. Melt curves were generated with the following parameters: 95°C for 15 seconds, 60°C for 1 minute, 95°C for 15 seconds, and 60°C for

15 seconds. Measurements were taken in biological triplicate with two technical replicates each. Relative transcript levels were calculated using the 2-ΔΔCt method [76]. Using GraphPad Prism software, P values were calculated using unpaired two-tailed Student's t-tests. The primers used for measuring RNA levels are listed in **S1 Table**.

## ChIP-qPCR and data analysis

ChIP was performed as described in [9] with some modifications. Buffers are listed in **S2 Table**. Briefly, ovaries from 200 1–2 day old adults were dissected in PBSP, homogenized with a pellet pestle, and crosslinked with 1.8% methanol-free formaldehyde (Thermo Fisher, cat# 28906) for 5 minutes. Fixation was quenched for 5 minutes by adding glycine to a final concentration of 225mM. The solution was removed, and the pellet was washed in PBSP 3 times. Samples were resuspended in 500μl PBSP and stored at -80˚C. Samples were thawed, centrifuged, and resuspended in 1ml lysis buffer 1. Samples were homogenized using sterile homogenizing beads in a Bullet Blender Homogenizer at 4˚C for six cycles of 30 seconds on and 1 minute off. Following centrifugation, cell pellets were washed at 4˚C for 10 minutes in lysis buffer 1, centrifuged, washed for 10 min at 4˚C in lysis buffer 2, centrifuged and then resuspended in 750μl lysis buffer 3. Chromatin was sheared to 100–500 base pairs at 4˚C with a Diagenode BioRuptor Pico for 15 cycles of 30 seconds on, and 30 seconds off.

For H3K9me3 ChIP, the chromatin lysates were precleared with a 1:1 mix of protein A/G Dynabead magnetic beads (Thermo Fisher, cat# 10001D and 10003D) for 1 hour at 4˚C and then incubated overnight at 4˚C with 1:1 mix of protein A/G Dynabead magnetic beads conjugated to H3K9me3 antibody (Abcam, cat# 8898 RRID: AB_306848).

For GFP ChIP, the chromatin lysates were precleared with Chromotek Magnetic Binding Control Agarose Beads (Proteintech, cat# bmab) for 1 hour at 4˚C, and then incubated overnight at 4˚C with Chromotek GFP-Trap Magnetic Agarose beads (Proteintech, cat# gtma).

Following immunoprecipitation, the samples were washed six times with ChIP-RIPA buffer, two times with ChIP-RIPA/500 buffer, two times with ChIP-LiCl buffer, and twice with TE buffer. DNA was eluted and reverse crosslinked from beads in 200μl elution buffer overnight at 65˚C. Following RNaseA and proteinase K treatment, DNA was recovered by phenol-chloroform extraction and used for qPCR.

qPCR was performed using *Power* SYBR Green PCR Master Mix (Thermo Fisher, cat# 4367659) with an Applied Biosystems QuantStudio 3 Real-Time PCR system. PCR steps were as follows: 95˚C for 10 minutes followed by 40 cycles of 95˚C for 15 seconds and 60˚C for 1 minute. Melt curves were generated with the following parameters: 95˚C for 15 seconds, 60˚C for 1 minute, 95˚C for 15 seconds, and 60˚C for 15 seconds. Primers as listed in **S1 Table**. ChIP experiments were performed on three independent biological samples. qPCR measurements on each sample were performed on at least two technical replicates. For H3K9me3 ChIP-qPCR, the ChIP to input enrichment (presented as percent input) is normalized to a negative control genomic region in the *rp49* locus. Statistical analysis was carried out by the GraphPad Prism software. The P values were calculated using unpaired two-tailed Student's t tests.

## Immunofluorescence and image analysis

Females were aged 3–5 days before gonad dissection. Ovaries were fixed and stained according to standard procedures with the following primary antibodies: mouse anti-*Drosophila* SXL (1:100, Developmental Studies Hybridoma Bank, cat# M18, RRID: AB_528464), rabbit anti-GFP (1:2500, Thermo Fisher, cat# A-11122, RRID: AB_221569) or FITC conjugated goat anti-GFP (1:750, Abcam, cat# ab6662, RRID: AB_305635), and rat anti-HA high affinity (1:500, Sigma, cat# 11867423001, RRID: AB_390919). The following secondary antibodies were used

at 1:200: Alexa Fluor 555 goat anti-rat (Thermo Fisher, cat# A-21434, RRID: AB_2535855), FITC goat anti-mouse (Jackson Immunoresearch Laboratories, cat# 115-095-003, RRID: AB_2338589), or FITC goat anti-rabbit (Jackson Immunoresearch Laboratories, cat# 111-095-003, RRID: AB_2337972). TO-PRO-3 Iodide carbocyanine monomer nucleic acid stain (1:1000, Thermo Fisher, cat# T3605) was used to stain DNA. Images were taken on a Leica SP8 confocal with 1024x1024 pixel dimensions, a scan speed of 600 Hz, and a frame average of 3. Sequential scanning was done for each channel, and three Z-stacks were combined for each image. Processed images were compiled with Microsoft PowerPoint.

## Analysis of publicly available RNA-seq data sets

The sources of the publicly available RNA-seq data generated from wild-type and mutant *Drosophila* tissues are listed in **S3 Table**. The data were downloaded from the SRA database at NCBI and analyzed using the tools available on the Galaxy web platform (https://usegalaxy.org/). FastQC assessed read quality and STAR was used to align the reads to the *Drosophila* reference genome (UCSC dm6). Screenshots of the expression data displayed on The Integrated Genome Viewer (IGV) are shown.

*D. simulans* and *D. yakuba phf7* orthologs were identified using the ortholog database at www.flybase.org. RNA-seq data from wild-type *D. simulans* and *D. yakuba* gonads was downloaded from the SRA database at NCBI (**S4 Table**). We uploaded the sequencing data to the Galaxy web platform (https://usegalaxy.org/), assessed read quality with FastQC and used STAR to align the reads to their respective reference genomes (listed in **S4 Table**). Browser screenshots of the expression data are displayed on the Integrated Genome Viewer (IGV).

## DNA sequence alignments

Multiple sequence alignments were performed in SnapGene (https://www.snapgene.com) using MUSCLE with default options. Pairwise sequence alignments were performed in BLASTn (https://blast.ncbi.nlm.nih.gov).

## Supporting information

**S1 Fig. The first intron of *phf7* contains seven copies of an ~250 bp DNA sequence.** Sequence alignment of the 7 ~250 bp sequences generated with the multiple sequence alignment tool MUSCLE embedded in the SnapGene software (snapgene.com).
(PDF)

**S2 Fig. Generation of the ΔR and ΔAR chromatin transgenic reporter lines.** ΔR transgene was designed to mimic the 5' end of the *phf7^ΔR* mutant gene, extending from the first male specific exon to the beginning of the open reading frame in exon 2. In the sequence below, the deleted R element is highlighted in purple. The ΔAR transgene contains a 2nd deletion, highlighted in yellow (region A). The sequences highlighted in blue and the primers located in region B). The constructs were generated by VectorBuilder's custom cloning services (https://en.vectorbuilder.com). The *phf7* fragments were inserted into their "user-defined promoter" modification of the pUASTattB expression vector. The transgenic constructs were sent to Rainbow Transgenic Flies Inc. for *phi-C31* catalyzed integration into the 65B2 PBac{y[+]-attP-3B}VK00033 site.
(PDF)

**S3 Fig. The first exon and a portion of the first intron are conserved between *D. melanogaster*, *D. simulans* and *D. yakuba*.** Pairwise alignments between the first exon and the adjacent intron were identified using the Basic Local Alignment Search Tool (BLASTn) available at

(https://blast.ncbi.nlm.nih.gov).
(PDF)

**S4 Fig. Reduced IDC-GFP staining in germ cells upon *idc* GLKD.** Confocal images of ovaries dissected from control (*nos>white-RNAi*) and mutant (*nos>idc-RNAi*) females carrying the IDC-GFP transgene and stained for GFP (green, white in A' and B') and DNA (red). Scale bar 50μm. **(A)** In control ovaries, IDC-GFP staining is observed in both the somatic cells and the germ cells. **(B)** In *idc* GLKD ovaries, no IDC-GFP staining is observed in the germ cells. As expected, staining is still observed in the somatic cells.
(PDF)

**S1 Table. Primers.**
(PDF)

**S2 Table. Buffers for ChIP.**
(PDF)

**S3 Table. The published RNA-seq data sets utilized in Fig 4.**
(PDF)

**S4 Table. The published RNA-seq data sets utilized in Fig 3.**
(PDF)

**S5 Table. Underlying numerical data for graphs in Figs 1, 2, 5 and 7.**
(XLSX)

## Acknowledgments

We would like to thank the members of the Cleveland fly community for helpful discussions, Jane Heatwole for fly food, and FlyBase as an essential resource. Stocks obtained from the Bloomington (NIH P40D018537) and The Vienna *Drosophila* Stock Centers were used in this study. Imaging was performed using equipment purchased through NIH Grant S10-OD016164, housed in the CWRU School of Medicine Imaging Core.

## Author Contributions

**Conceptualization:** Helen K. Salz.

**Data curation:** Laura Shapiro-Kulnane, Micah Selengut, Helen K. Salz.

**Formal analysis:** Laura Shapiro-Kulnane, Micah Selengut, Helen K. Salz.

**Funding acquisition:** Helen K. Salz.

**Investigation:** Laura Shapiro-Kulnane, Micah Selengut, Helen K. Salz.

**Supervision:** Helen K. Salz.

**Writing – original draft:** Helen K. Salz.

**Writing – review & editing:** Laura Shapiro-Kulnane, Micah Selengut, Helen K. Salz.

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
