## [Decision Letter · Decision Letter 0]

20 Nov 2022

Dear Helen,

Thank you very much for submitting your Research Article entitled 'Safeguarding Drosophila female germ cell identity depends on an H3K9me3 mini domain guided by a ZAD zinc finger protein' to PLOS Genetics.

The manuscript was fully evaluated at the editorial level and by independent peer reviewers. The reviewers appreciated the attention to an important topic but identified some concerns that we ask you address in a revised manuscript.

We therefore ask you to modify the manuscript according to the review recommendations. Specifically, reviewer #1 has excellent suggestions that can be addressed by text/figure changes and would improve the clarity of the manuscript. As noted by reviewer #2 IDC RNAi knockdown needs validation; please clarify IDC expression pattern and timing, especially in the germline; please clarify where IDC binds, and what regions may be sufficient, as commented in point #5 by reviewer 2. To address this question of sufficiency, you may wish to add new data, however new data are not required. Additionally, reviewer #3 "major comment" #2 asks for validation of IDC RNAi (also noted by reviewer 2). Please use a second RNAi line, if available, or confirm knock-down via qPCR and/or some kind of protein quantification (e.g. western or immunofluorescence). If another RNAi line is not available, then it should be clearly stated how possible off-target effects of the one line you used is mitigated or otherwise addressed, including the possibility that off-target effects may not be completely excluded. It seems, with the exception of RNAi line validation, that all other major and minor points raised by each of the reviewers can be addressed by addition of data you already have (i.e. no new experiments; for example, quantification of IDC-GFP where possible during development of images you already have), text changes, and revising or reorganizing figures and legends, as suggested by the reviewers. As noted below, please make your best effort to address all reviewer comments in your cover letter, and by making minor typo corrections and figure revisions in your revised manuscript.

Yours sincerely,

Giovanni Bosco, Ph.D.

Academic Editor

PLOS Genetics

John Greally

Section Editor

PLOS Genetics

Reviewer's Responses to Questions

**Comments to the Authors:**

Reviewer #1: This is a well-written paper with a series of logical, well executed experiments that support the author’s conclusion that the ZAD zinc-finger protein IDC plays a direct role in establishing the H3K9me3 domain that regulates the expression of the phf7 gene in the ovary. The importance of this finding is well described in the manuscript: while a lot is known about how H3K27me3-repressive chromatin domains are established, little is known about how small H3K9me3 domains are established. Making good use of public data, the authors clearly show that the piRNA pathway, important in forming large H3K9me3 domains, is not required for the formation of the small H3K9me3 domain that covers the male-specific phf7 promoter. The authors identified the ZAD zinc-finger gene IDC in a previous RNAi screen as a candidate for a phf7 transcriptional regulator. Here they show that (1) knocking down idc with RNAi in ovaries leads to the production of the male-specific phf7 RNA and production of the phf7 protein in ovaries 2) H3K9me3 levels are reduced over the phf7 gene in idc germline mutant clones and (3) IDC binds to the phf7 gene. This paper is made even more interesting by the potential similarities between the recruitment of the H3K9me3 methyltransferase by the KRAB-zinc finger family in mammals and the ZAD zinc-finger family in flies.

I have a few questions:

1) Regarding Fig. 1A and the discussion of the repetitive element: Does the phf7 gene from the related Drosophila species also have a repetitive element (perhaps unrelated)?

2) Regarding Fig. 2A. It’s not clear to me exactly what is in the transgene. From the diagram it looks like TSS1 and part of the non-coding region upstream of TSS2 are in the transgene, however, the title of the figure is “Non-coding sequences within the first intron are sufficient for H3K9me3 deposition.” This is important because IDC binds to TSS1. If the exons are included then change the title to “Non-coding sequences within the first intron are required for H3K9me3 deposition”

3) Fig. 3A, please put a diagram of the vector used to do this experiment. Where were the fragments cloned in? I can’t really visualize it based on the description in the methods.

4) Fig. 3B would be more informative if the rest of the gene was also shown (same as Fig. 1A). It says in the text that the rest of the DNA is not conserved—show this in the figure. Also, please indicate how many bases are conserved, 84% in a region of how many nucleotides, etc. I see the sequence conservation in Fig. S2, just state the number of bases figure 3A and perhaps state that the sequence conservation is in Fig. S2 in the figure legend.

5) Pg. 11, line 258. “our studies provide one of the first examples of a ZAD-ANF protein guiding H3K9me-mediated gene silencing.” What are the other examples? Please reference.

Reviewer #2: In this manuscript, Shapiro-Kulnane et al. take advantage of the phf7 locus to investigate how H3K9me3 mediated silencing is promoted at protein coding genes. This builds on prior work from the lab that had identified H3K9me3 deposition over a testis-specific transcription start site for phf7 as important for silencing expression of the male-specific protein. In a recently published screen, the authors identified CG4936 (IDC) as a putative regulator of phf7 expression. Here, they link these two studies by directly showing that IDC binds near the male-specific TSS of phf7 and is essential for H3K9me3. Overall the manuscript is clear, and the experiments are rigorously performed. However, the impact beyond what has been previously shown by the lab is somewhat limited and additional experiments could strengthen the conclusions and the model.

Major issues:

1. The major novel finding in this work is the role of IDC in transcriptional repression in the germline, which was already hinted at by the results of the RNAi screen. To further support this conclusion, the authors should validate the specificity of their RNAi construct and/or use an orthogonal approach to confirm the specificity.

2. The expression pattern of IDC as a whole is not clearly explained. Is IDC expressed broadly in all tissues or only in the germline? In the Discussion, the authors mention that IDC is also expressed in male germ cells, suggesting that the simple model presented in Figure 7B is misleading. The authors have all the tools in hand to test if IDC binds phf7 in the male germline. The authors discuss a potential candidate for mediating the female-specific functions (STWL). It is unclear why they do not directly test this hypothesis.

3. To better understand the function of IDC in the female-specific silencing, it would be useful to determine IDC binding beyond the phf7 locus both in the male and female germlines. This would provide insights into additional targets and possibly more generally into the role of the ZAD-ZNF gene family. Along these same lines, in Figure 6 IDC-GFP appears to be broadly expressed in the female germline. It would be useful for the authors to test and/or discuss in which cell types IDC is required. Does overexpression of IDC lead to H3K9me3 deposition over phf7?

4. A major conclusion from the manuscript is that the determinants required for H3K9me3 deposition at phf7 are within element A of the 1st intron. However, additional experiments would strengthen this conclusion, which is currently based on the fact that deletion of a set of repeats does not disrupt H3K9me3 while a deletion that also includes the A region does. With the data presented, it remains possible that these two regions are redundant. Deletions of the A region alone should be tested. Likewise, it would be useful to test sufficiency. Does insertion of the A region in a transgene lead to H3K9me3 in the female germline?

5. As written some of the data seems contradictory. The A region in the first intron appears important for H3K9me3 deposition, but IDC appears to bind the exon (Figure 7A). Do the authors think that there are additional factors that recognize the A region? At the very least, the authors should discuss these conflicting data.

Minor issues:

In line 125 it says the p value is 0.0003, however in the corresponding figure it says the p value is 0.003

All of the comparisons made with RT-qPCR and ChIP-qPCR should have statistics.

Figure 4 should include scales for the y-axes for the genome browser tracks.

Reviewer #3: Heterochromatin is guided to transposable elements (TEs) to silence their transcription. Sequence-specific transcription factors such as KRAB-ZFPs in mammals and ZAD-ZNF in the fly have been shown to guide this heterochromatin formation. However, how protein-coding genes are targeted for silencing is not fully understood. During, Drosophila oogenesis testis-specific version of PHD finger protein 7 (phf7) transcription is silenced. Using lineage-specific transcription of ph7 as a paradigm, the authors find that member of the ZAD-ZNF protein family that they name Identity Crisis (IDC) is necessary for H3K9me3 deposition in a sequence-specific manner on phf7 gene locus. In this study, the authors specifically claim that:

1. Conserved sequences in the first intron are required for H3K9me3 deposition

2. IDC is expressed in the undifferentiated cells of the germline.

3. IDC silences testis-specific ph7 expression through H3K9me3 deposition dependent on SETDB1 but independent of piRNA machinery.

4. IDC binds to phf7 locus to directly promote silencing of phf7 locus

Major Comments:

1. Figure 1 is superfluous with Figure 2 and can be combined. It was confusing to read that something is not required for first.

2. IDC RNAi data needs additional validation. The authors have only used one RNAi line. They need to test another RNAi line or mutant clones.

3. The phenotype of IDC depletion should be described in more detail and compared to SETDB1 mutants.

4. Does the loss of IDC lead to depletion of H3K9me3 at a global level?

Minor Comments:

1. Figure 3B- the lines in the table are not perpendicular

2. Y axis for Figure 4 is needed (X TPMs)

3. Figure 5A requires significance

4. Authors should add grayscale for staining where possible

5. IDC-GFP quantification of levels as a function of development in the germarium is needed.

**Have all data underlying the figures and results presented in the manuscript been provided?**

Reviewer #1: Yes

Reviewer #2: Yes

Reviewer #3: Yes

PLOS authors have the option to publish the peer review history of their article (what does this mean?). If published, this will include your full peer review and any attached files.

Reviewer #1: No

Reviewer #2: No

Reviewer #3: No

---

## [Editor Report · Decision Letter 1]

12 Dec 2022

Dear Dr Salz,

We are pleased to inform you that your manuscript entitled "Safeguarding Drosophila female germ cell identity depends on an H3K9me3 mini domain guided by a ZAD zinc finger protein" has been editorially accepted for publication in PLOS Genetics. Congratulations!

Yours sincerely,

Giovanni Bosco, Ph.D.

Academic Editor

PLOS Genetics

John Greally

Section Editor

PLOS Genetics

Comments from the reviewers (if applicable):

**Data Deposition**

http://datadryad.org/submit?journalID=pgenetics&manu=PGENETICS-D-22-01219R1

**Press Queries**

---

## [Editor Report · Acceptance letter]

16 Dec 2022

PGENETICS-D-22-01219R1 

Safeguarding Drosophila female germ cell identity depends on an H3K9me3 mini domain guided by a ZAD zinc finger protein 

Dear Dr Salz, 

We are pleased to inform you that your manuscript entitled "Safeguarding Drosophila female germ cell identity depends on an H3K9me3 mini domain guided by a ZAD zinc finger protein" has been formally accepted for publication in PLOS Genetics! Your manuscript is now with our production department and you will be notified of the publication date in due course.

With kind regards,

Anita Estes

PLOS Genetics

On behalf of:
